# Effects of Lactoferrin on Prevention of Acute Gastrointestinal Symptoms in Winter: A Randomized, Double-Blinded, Placebo-Controlled Trial for Staff of Kindergartens and Nursery Schools in Japan

**DOI:** 10.3390/ijerph17249582

**Published:** 2020-12-21

**Authors:** Masaru Mizuki, Teruomi Tsukahara, Hirotsugu Oda, Miyuki Tanaka, Koji Yamauchi, Fumiaki Abe, Tetsuo Nomiyama

**Affiliations:** 1Department of Occupational Medicine, School of Medicine, Shinshu University, 3-1-1 Asahi, Matsumoto, Nagano 390-8621, Japan; mizuki@shinshu-u.ac.jp (M.M.); nomiyama@shinshu-u.ac.jp (T.N.); 2Department of Preventive Medicine and Public Health, School of Medicine, Shinshu University, 3-1-1 Asahi, Matsumoto, Nagano 390-8621, Japan; 3Food Ingredients and Technology Institute, R&D Division, Morinaga Milk Industry Co., Ltd., Zama, Kanagawa 252-8583, Japan; h-oda@morinagamilk.co.jp (H.O.); m_tanaka@morinagamilk.co.jp (M.T.); ko_yamau@morinagamilk.co.jp (K.Y.); f_abe@morinagamilk.co.jp (F.A.)

**Keywords:** lactoferrin, diarrhea, kindergarten, winter, subjective acute gastrointestinal symptoms

## Abstract

This study investigated the preventive effects of lactoferrin (LF) on subjective acute gastrointestinal symptoms during the winter in a randomized, double-blinded, placebo-controlled parallel-group comparative trial. The eligible subjects were healthy adults working at kindergartens and nursery schools. We randomized the subjects to the Placebo group (0 mg/day), the Low LF group (200 mg/day), and the High LF group (600 mg/day) for 12 weeks. The prevalence of acute gastrointestinal symptoms was significantly lower in the High LF (13/112 vs. 26/116; *p* = 0.030) and the Low LF (13/107 vs. 26/116; *p* = 0.040) groups than in the Placebo group. The adjusted odds ratio for the prevalence of acute gastrointestinal symptoms was 2.78 (95% CI: 1.19–6.47) in the Placebo group compared with the High LF group. LF is useful to prevent acute gastrointestinal symptoms among childcare workers, who mainly consist of women.

## 1. Introduction

Infectious diseases such as infectious gastroenteritis are prevalent in the winter, and the prevention of these diseases is a public health issue in Japan [1,2,3]. Children in kindergartens and nursery schools have a particularly high risk of infection due to their suppressed and immature immune systems [4,5,6] and their school environments [7]. Childcare workers also have a high risk of infection in the same environment.

These infectious diseases are mainly caused by viruses. Noroviruses are a leading cause of infectious gastroenteritis in the winter, and rotaviruses, sapoviruses, and astroviruses can also cause gastroenteritis [8]. Infectious gastroenteritis is a contagious disease across all age groups, and it causes symptoms such as fever, abdominal pain, nausea, vomiting, and diarrhea [9]. As the genome of noroviruses easily mutates, no vaccines are currently available [10].

Lactoferrin (LF) is an iron-binding glycoprotein found in the milk of most mammals [11]. In vitro studies demonstrated LF to have antiviral effects against norovirus [12], rotavirus [13], and norovirus surrogates [12,14]. LF also has immune stimulatory effects; it increased the production of type I interferon (IFN), which inhibits the replication of viruses in the Peyer’s patches in the small intestine, and activated natural killer (NK) cells, which are cytotoxic to virally infected cells [15]. Furthermore, it caused isotype switching of B cells and increased the production of secretory immunoglobulin A in the small intestine, which blocks the attachment of pathogens to mucosa [16]. LF also activates CD4+ and CD8+ T cells in lymphoid tissues in the small intestine [17].

Formula, yogurt, supplements, and other food products are fortified with LF derived from bovine milk, and the consumption of these food products is expected to help prevent infectious diseases [18]. We conducted the present study to assess whether the intake of LF has preventive effects against subjective acute gastrointestinal symptoms in the healthy adult staff of kindergartens and nursery schools.

## 2. Materials and Methods

### 2.1. Study Design and Subjects

This study was a randomized (1:1:1), double-blinded, placebo-controlled parallel-group comparative trial conducted at the Department of Preventive Medicine and Public Health, Shinshu University School of Medicine, Nagano Prefecture, Japan. We conducted this trial, according to the current revision of the Declaration of Helsinki (2013) and the Ethical Guidelines for Medical and Health Research Involving Human Subjects (2015) [19]. The protocol and informed consent form were approved by the Institutional Review Board (IRB) at Shinshu University School of Medicine on 4 November 2015. This trial was registered in the University Hospital Medical Information Network (UMIN) Clinical Trials Registry in Japan on 11 November 2015 (registration no. UMIN000019752). Full details of the trial protocol (in Japanese) are available from the corresponding author on reasonable request.

Eligible subjects were healthy adults aged ≥20 years working at a kindergarten or nursery school in Nagano Prefecture, Japan. The exclusion criteria were milk allergy, pregnancy, plans to resign from the staff position, history of serious disorders of the liver, kidney, heart, lung, gastro-intestine, blood, endocrine system, or metabolic system, habitual consumption of LF, and being judged as inappropriate for this trial by a principal investigator (e.g., due to lactation). We stratified the subjects by kindergartens or nursery schools and then further randomized them into three blocks (1:1:1 ratio).

An allocation manager unrelated to this trial used computer-generated lists of random numbers, prepared an allocation table, and consecutively numbered the test foods in accordance with that table. He concealed the allocation table from the investigators, sealed it in an opaque envelope, and kept it until the code was broken at the end of the study. The investigators and subjects were blinded during this period. Investigators enrolled participants, assigned the order number, and sent the corresponding number of test foods to the subjects.

### 2.2. Intervention

The trial period was from November 2015 to March 2016, selected because outbreaks of gastroenteritis tend to occur during the winter. During the intervention period, the subjects were instructed to swallow six tablets per day (containing either 0, 200, or 600 mg of LF) with water at bedtime. The tablets were round, reddish-orange tablets (250 mg, 9.1 mm DIA., and 4.8 mm thick). The tablets taken by the High LF group contained 100 mg of bovine LF (purity ≥96% by HPLC) derived from cheese whey (identical to the commercial product “Lactoferrin Original”, Morinaga Milk Industry Co., Tokyo, Japan). The tablets taken by the Low LF group contained 33.3 mg of LF and those taken by the Placebo group contained 0 mg of LF. Morinaga Milk Industry provided research funding and test foods for this trial.

The allocation manager evaluated the appearance, solidness, smell, taste, etc. of the tablets, and the bottles and cartons of the three types of test foods at both the allocation and code breaking, and confirmed that they were indistinguishable. Code breaking was performed after the database and statistical analysis plan were locked. The subjects recorded their intake of tablets, and their acute gastrointestinal symptoms (fever ≥37 °C, abdominal pain, nausea, vomiting, and diarrhea) based on a diagnosis and subjective acute gastrointestinal symptoms. They recorded other physical changes using self-completed questionnaires.

### 2.3. Endpoints

The primary endpoint of this study was the prevalence of acute gastrointestinal symptoms. The number of subjects who recorded acute gastrointestinal symptoms was compared. Secondary endpoints were the episodes and average duration of acute gastrointestinal symptoms. One episode was defined as a block of consecutive days and the average duration was defined as the average number of consecutive days of the total episodes. Cumulative prevalence days of subjective acute gastrointestinal symptoms were also compared to evaluate the prevalence, episode, and duration comprehensively. They were calculated as the sum of the days of the symptoms.

Any unfavorable and unintended sign, symptom, or disease temporarily associated with the intake of test foods was defined as an adverse event. The adverse events were evaluated by referring to the Revised National Cancer Institute–Common Toxicity Criteria (NCI-CTCAE) ver. 4.0. Any adverse event that exhibited a causal relationship with the intake of a test food that was unable to be excluded was defined as an adverse drug reaction.

### 2.4. Sample Size and Statistical Analysis

The prevalence of acute gastrointestinal symptoms was estimated to be 20%. The expected intake of LF was estimated to reduce the prevalence to 6.7% (one-third) with a type 1 error of 0.05 and a power of 80%. The estimated necessary sample size was 306 (102 in each group) with an estimated dropout rate of 33% (one-third), and a target sample was set as 450 (150 in each group). In accordance with the closed testing procedure, the χ^2^ test or Fisher’s exact test was used to analyze the prevalence of acute gastrointestinal symptoms and cumulative prevalence days of subjective symptoms, and the Mann–Whitney U test was used to analyze the episodes and average duration of acute gastrointestinal symptoms. The χ^2^ test for trend and the Jonckheere–Terpstra trend test were used to analyze trends. Probability values <0.05 were considered significant. For the exploratory analysis, we used data with an intake rate of ≥70% (261 subjects: Placebo, *n* = 84; Low LF, *n* = 85; High LF, *n* = 92). Multiple logistic regression analysis was used to determine the odds ratios with a 95% confidence interval (95% CI) for the associations between the prevalence of acute gastrointestinal symptoms as a dependent variable and the independent variables of the participant’s age and LF intake in the exploratory analysis. All analyses were conducted using the Statistical Package for Social Sciences (SPSS) ver. 24.0 (SPSS, Chicago, IL, USA).

## 3. Results

Subjects were recruited between 12 November and 17 December, 2015 (35 days). The intervention period was from 21 December 2015 to 13 March 2016 (12 weeks). The post-intervention period was from 14 March 2016 to 31 March 2016 (18 days). We invited 67 kindergartens and nursery schools (828 staff members) in Matsumoto-city, Ikeda-town, Minowa-town, Yamagata-village, and Chikuhoku-village in Nagano Prefecture, Japan to participate. The directors of 56 facilities (778 staff members) agreed to do so. We visited 28 facilities (378 staff members) and explained the details of the trial and informed consent form. As an alternative method, we sent video media (DVD or YouTube URL) with informed consent forms to 28 facilities (400 staff members). Although 350 childcare workers consented to participate, four were excluded (three for habitual consumption of LF and one who was determined to be inappropriate for this study). After randomization, nine subjects retracted their consent and two did not submit questionnaires. A final total of 335 staff members were analyzed: Placebo, *n* = 116; Low LF, *n* = 107; High LF, *n* = 112. A full analysis set (FAS) of the data of 335 subjects was used in the intervention period for the primary analyses (Appendix A). For the exploratory analysis, data with an intake rate of ≥70% (261 subjects: Placebo, *n* = 84; Low LF, *n* = 85; High LF, *n* = 92) and data collected during the post-intervention period (280 subjects) were used. The average LF intake rates were not significantly different among the three groups (Table 1).

The prevalence of acute gastrointestinal symptoms showed a significant trend, and was significantly lower in the High LF and Low LF groups than in the Placebo group. The average duration of diarrhea showed a significant trend, and was significantly shorter in the High LF group than in the Placebo group (Table 2). The cumulative prevalence days of abdominal pain, nausea, diarrhea, and fever showed a significant trend, and those of abdominal pain, diarrhea, and fever were significantly fewer in the High LF group and Low LF group than in the Placebo group. Those of nausea were significantly fewer in the High LF group than in the Placebo group (Appendix A).

As the only data with an intake rate of ≥70% were used for the exploratory analysis, 32, 22, and 20 subjects in the Placebo, Low LF, and High LF groups, respectively, were excluded from the FAS (Table 3). The average intake rate was 90.9% in the Placebo group, 91.5% in the Low LF group, and 91.4% in the High LF group, with no significant difference (Table 1). The prevalence and average duration of acute gastrointestinal symptoms showed a significant trend, and were significantly lower in the High LF group than in the Placebo group. The average duration of diarrhea showed a significant trend, and was significantly shorter in the High LF group than in the Placebo group. The prevalence of nausea showed a significant trend. The average duration of fever also showed a significant trend.

The relationships between the prevalence of acute gastrointestinal symptoms and LF intake were calculated using the logistic regression analysis. Adjusted odds ratios (ORs) were calculated after adjusting for age and LF intake as independent variables. We used data with an intake rate of ≥ 70% and excluded 13 men because the absentees were only women. The adjusted OR for the prevalence of acute gastrointestinal symptoms was 2.78 (95% CI: 1.19–6.47) in the Placebo group compared with the High LF group (Table 4).

Regarding the post-intervention period for exploratory analysis, 16, 18, and 16 subjects in the Placebo, Low LF, and High LF groups, respectively, were excluded because they did not return a questionnaire. Data from one Placebo subject and four High LF subjects were excluded from the FAS due to their intake of remaining test foods. The prevalence of acute gastrointestinal symptoms in the post-intervention period (2 weeks) was 5/99 (5.1%) in the placebo group, 2/89 (2.2%) in the Low LF group, and 2/92 (2.2%) in the High LF group, and no significant difference was observed among the three groups.

The number of subjects with adverse events was 58/116 (50%) in the Placebo group, 68/107 (63.6%) in the Low LF group, and 64/112 (57.1%) in the High LF group, with no significant difference among the groups (*p* = 0.124; Fisher’s exact test). Major symptoms were hay fever, menstrual pain, and fatigue. None of these symptoms had a causal relationship with the intake of test foods.

## 4. Discussion

This study demonstrated that the intake of LF is useful to prevent acute gastrointestinal symptoms in childcare workers. Based on the FAS, the prevalence of acute gastrointestinal symptoms showed a significant trend, and was significantly lower in the High LF and Low LF groups than in the Placebo group. The average duration of diarrhea showed a significant trend, and was significantly shorter in the High LF group than in the Placebo group. In addition, the cumulative prevalence days of abdominal pain, nausea, diarrhea, and fever showed a significant trend, and those of abdominal pain, diarrhea, and fever were significantly fewer in the High LF group and Low LF group than in the Placebo group. Those of nausea were significantly fewer in the High LF group than in the Placebo group. Based on data with an intake rate of ≥70%, the prevalence and average duration of acute gastrointestinal symptoms showed a significant trend, and were both significantly lower in the High LF group than in the Placebo group. The average duration of diarrhea showed a significant trend, and was significantly shorter in the High LF group than in the Placebo group. The prevalence of nausea and average duration of fever showed a significant trend. The adjusted OR for the prevalence of acute gastrointestinal symptoms was 2.78 (95% CI: 1.19–6.47) in the Placebo group compared with the High LF group. Thus, the intake of LF helps to reduce the prevalence of acute gastrointestinal symptoms.

In previous trials, 36 mg/day of LF reduced the incidence and duration of diarrhea-related illness in infants [20]. The intake of LF (48 mg/day)-fortified formula reduced the prevalence of acute gastrointestinal symptoms in children aged 12–32 months [21]. Furthermore, 1 g/day of LF alleviated symptoms of diarrhea in children [22]. The consumption of LF-containing yogurt (100 mg/day) for ≥3 days/week may help to alleviate vomiting among children in nursery schools [6]. The intake of 100 mg/day of LF with prebiotics and probiotics alleviated symptoms of rotavirus gastroenteritis in children, and reduced the incidence of norovirus gastroenteritis in adults [23,24]. Therefore, at least 100 mg/day of LF may be required for the preventive effects against gastroenteritis in children. A higher dosage or intake rate of LF may be more effective and alleviate the acute gastrointestinal symptoms in adults. However, there is no evidence regarding the suppressive effects of LF alone on acute gastrointestinal symptoms in adults, except for the suppressive effects of 400 mg/day of LF on Helicobacter pylori infection [25]. In our study, 200 and 600 mg/day of LF reduced the prevalence of acute gastrointestinal symptoms. Furthermore, 600 mg/day of LF (intake rate more than 70%) shortened the average duration of acute gastrointestinal symptoms and diarrhea, and showed a significant trend in gastroenteritis, diarrhea, and fever. The present study is the first to demonstrate the preventive effects of LF alone (200 and 600 mg/day) on acute gastrointestinal symptoms in adults.

A major pathogen of infectious gastroenteritis in winter is norovirus [8]. Norovirus enters the mouth, survives the gastric fluid, reaches the small intestine, and infects epithelial cells and immune cells in Peyer’s patches [26]. In contrast, some orally ingested LF is hydrolyzed by the gastric fluid, and a mixture of LF and its peptides reaches the small intestine [27,28]. LF binds to epithelia in the bovine small intestine and preferentially binds to epithelia overlaying Peyer’s patches [29]. In vitro studies suggested that LF and its peptide inhibit the binding of norovirus surrogates (feline calicivirus and murine norovirus) to their target cells and suppress infection [12,14].

Interferon (IFN) -α, β, and λ are antiviral cytokines that inhibit the replication of norovirus [30,31]. The administration of LF to mice promoted the production of IFN-α and IFN-β in epithelia and Peyer’s patches in the small intestine [15,32]. LF increased the production of IFN-λ in intestinal epithelial cells [33]. Furthermore, LF increased the production of IFN-β and reduced human norovirus replication in a B cell culture model [34]. It also inhibited the replication of murine norovirus via the production of IFN-α and -β [12]. Therefore, ingested LF and its peptides may inhibit the binding and replication of norovirus in epithelial cells and immune cells in Peyer’s patches in the small intestine, thereby suppressing infectious gastroenteritis.

In the post-intervention period, although short, we observed no significant difference in the prevalence or duration of acute gastrointestinal symptoms. As ingested LF goes through the digestive tract, direct effects on pathogens and the host cannot be expected during the post-intervention period. In addition, immune stimulatory effects of LF disappear within 3 weeks after its consumption [35]. The continuous intake of LF is therefore required to maintain its preventive effects against infectious diseases.

The prevalence rate of acute gastrointestinal symptoms in our trial was 15.5%. In epidemiological surveillance of infectious diseases in Nagano Prefecture, the incidence of infectious gastroenteritis in the intervention period was 4496. As the population of Nagano Prefecture is ~2.1 million [36], the incidence rate of infectious gastroenteritis was calculated to be approximately 0.2%; these values are markedly lower than those in our present trial. However, almost all of these rates were reported from pediatric clinics, and there are few studies on the incidence rates in adults and in relation to home remedies. It is therefore difficult to estimate the precise incidence rates in adults and compare the reported rates with ours. As epidemic outbreaks of infectious diseases occur easily among children in nursery schools and kindergartens, staff members of these facilities have a higher-than-average risk of infection compared with adults who do not work in those environments, and the prevalence rates in our trial may reflect this environmental factor.

These are several limitations in this study. First, the majority of the subjects were female childcare workers and the small sample size of male subjects might make it difficult to generalize our results. Second, the subjects reported the results of diagnoses and symptoms based on their subjective judgement. Their reports may have directly affected the study endpoint. Third, although the trial was a randomized, double-blinded, placebo-controlled parallel-group comparative trial, we were not able to consider other healthy habits in our analyses. Other healthy habits may have affected the study endpoints. Further trials considering the subjects’ occupation, male/female ratio, and diagnostic methods for infectious diseases are required.

## 5. Conclusions

LF is useful for the prevention of acute gastrointestinal symptoms among childcare workers, who mainly consist of women.

## Figures and Tables

**Table 1 ijerph-17-09582-t001:** Baseline demographic and clinical characteristics.

		All Subjects					Intake Rate Greater than 70%			
	Total	Placebo	Low LF	High LF	*p*-Value	Total	Placebo	Low LF	High LF	*p*-Value
Subjects, *n*	335	116	107	112		261	84	85	92	
Sex, *n*										
Men	16	7	7	2	0.168	13	6	6	1	0.070
Women	319	109	100	110		248	78	79	91	
Average age ± SD, yr	40.6 ± 13.2	40.3 ± 13.3	41.2 ± 13.7	40.2 ± 12.7	0.850	42.6 ± 12.8	42.0 ± 13.1	43.5 ± 13.2	42.4 ± 12.2	0.828
Intake rate, %	81.5 ± 20.8	79.4 ± 20.0	82.7 ± 20.3	82.7 ± 22.1	0.061	90.9 ± 8.8	89.8 ± 8.9	91.5 ± 8.6	91.4 ± 9.0	0.197

SD: standard deviation.

**Table 2 ijerph-17-09582-t002:** Prevalence, episodes, and average duration of acute gastrointestinal symptoms.

	Placebo	Low LF ^1^	High LF ^2^	*p*-Value ^1^	*p*-Value ^2^	*p* for Trend
Subjects, *n*	116	107	112			
Acute gastrointestinal symptoms						
Number, *n*	26	13	13	0.044	0.030	0.024
Prevalence, %	22.4	12.1	11.6			
Episodes, mean (SD)	1.3 (0.7)	1.4 (0.5)	1.2 (0.4)	0.282	0.704	0.991
Average duration, mean (SD)	2.5 (1.6)	2.2 (1.1)	1.8 (1.1)	0.748	0.166	0.159
Abdominal pain						
Number, *n*	16	10	4			
Prevalence, %	61.5	76.9	30.8	0.337	0.096	0.131
Average duration, mean (SD)	2.3 (1.8)	1.9 (1.0)	2.3 (1.3)	0.762	0.766	0.933
Nausea						
Number, *n*	15	9	4			
Prevalence, %	57.7	69.2	30.8	0.728	0.176	0.184
Average duration, mean (SD)	1.9 (1.4)	1.9 (1.0)	2.3 (1.0)	0.700	0.258	0.347
Vomiting						
Number, *n*	9	3	5			
Prevalence, %	34.6	23.1	38.5	0.714	0.813	0.930
Average duration, mean (SD)	1.2 (0.4)	1.3 (0.6)	1.2 (0.4)	0.712	0.925	1.000
Diarrhea						
Number, *n*	13	9	8			
Prevalence, %	50.0	69.2	61.5	0.318	0.496	0.402
Average duration, mean (SD)	2.0 (1.5)	1.6 (0.5)	1.0 (0.0)	1.000	0.029	0.036
Fever						
Number, *n*	11	7	4			
Prevalence, %	42.3	53.8	30.8	0.496	0.728	0.615
Average duration, mean (SD)	1.8 (0.9)	1.1 (0.4)	1.3 (0.5)	0.077	0.250	0.087

^1^: placebo vs. low LF, ^2^: placebo vs. high LF, SD: standard deviation. The unit of average duration is days.

**Table 3 ijerph-17-09582-t003:** Prevalence, episodes, and average duration of acute gastrointestinal symptoms (intake rate of ≥ 70%).

	Placebo	Low LF ^1^	High LF ^2^	*p*-Value ^1^	*p*-Value ^2^	*p* for Trend
Subjects, *n*	84	85	92			
Acute gastrointestinal symptoms						
Number, *n*	19	11	10			
Prevalence, %	22.6	12.9	10.9	0.100	0.036	0.033
Episodes, mean (SD)	1.3 (0.6)	1.3 (0.5)	1.2 (0.4)	0.770	0.896	0.972
Average duration, mean (SD)	2.7 (1.8)	2.3 (1.2)	1.3 (0.7)	0.843	0.021	0.025
Abdominal pain						
Number, *n*	10	8	2			
Prevalence, %	52.6	72.7	20.0	0.442	0.126	0.183
Average duration, mean (SD)	2.7 (2.2)	1.9 (1.1)	1.5 (0.7)	0.963	0.646	0.762
Nausea						
Number, *n*	12	7	2			
Prevalence, %	63.2	63.6	20.0	1.000	0.050	0.045
Average duration, mean (SD)	2.0 (1.5)	2.1 (1.0)	1.5 (0.7)	0.478	0.841	0.743
Vomiting						
Number, *n*	6	2	2			
Prevalence, %	31.6	18.2	20.0	0.672	0.675	0.442
Average duration, mean (SD)	1.2 (0.4)	1.5 (0.7)	1.0 (0.0)	0.378	0.564	1.000
Diarrhea						
Number, *n*	9	7	7			
Prevalence, %	47.4	63.6	70.0	0.466	0.433	0.222
Average duration, mean (SD)	2.3 (1.7)	1.6 (0.4)	1.0 (0.0)	0.619	0.024	0.013
Fever						
Number, *n*	8	6	3			
Prevalence, %	42.1	54.5	30.0	0.510	0.694	0.651
Average duration, mean (SD)	2.0 (0.9)	1.2 (0.4)	1.0 (0.0)	0.071	0.091	0.022

^1^: placebo vs. low LF, ^2^: placebo vs. high LF, SD: standard deviation. The unit of average duration is days.

**Table 4 ijerph-17-09582-t004:** Relationship between the prevalence of acute gastrointestinal symptoms and lactoferrin (LF) intake by logistic regression analysis.

Variables	*n*	Crude Odds Ratio			Adjusted Odds Ratio		
		Odds Ratio	95% CI	*p*-Value	Odds Ratio	95% CI	*p*-Value
Age							
<43 years old	118	1.61	0.81–3.19	0.173	1.75	0.87–3.51	0.116
≥43 years old	130	reference			reference		
LF intake			-		-		
Placebo group	78	2.61	1.13–6.02	0.025	2.78	1.19–6.47	0.018
Low LF group	79	1.31	0.53–3.27	0.563	1.34	0.53–3.36	0.533
High LF group	91	reference	-		reference	-	

Adjusted odds ratio: Adjusted for age and LF intake.

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
