# Peer review of "Effects of Lactoferrin on Prevention of Acute Gastrointestinal Symptoms in Winter: A Randomized, Double-Blinded, Placebo-Controlled Trial for Staff of Kindergartens and Nursery Schools in Japan"

_ijerph, 2020, doi:10.3390/ijerph17249582_

Round 1

Reviewer 1 Report

Reviewers' comments:

This manuscript by dr. Mizuki and co-workers “Effects of lactoferrin on prevention of infectious gastroenteritis in winter: A randomized, double-blinded, placebo-controlled trial for staff of kindergartens and nursery schools in Japan” examines the preventive effect of bovine lactoferrin on infectious gastroenteritis in childcare workers. The study was carried out in 335 healthy workers from 28 kindergartens and nursery schools during the winter season.

The authors stratified the subjects by kindergartens or preschools and then further randomized them into three groups (1:1:1 ratio): the workers who would not be given lactoferrin (n=116, Placebo group); the workers who would be given 200 mg/day lactoferrin for 12 weeks (n=107, Low LF group) and the workers who would be given 600 mg/day lactoferrin for 12 weeks (n=112, High LF group). For the exploratory analysis were utilised data with an intake rate of ≥70% (261 subjects: Placebo, n=84; Low LF, n=85 High LF, n=92) and data collected during the post-intervention period (18 days, 280 subjects).

The results of this study demonstrate for the first time the preventive effects of lactoferrin alone on acute gastroenteritis in adults.

General Comments:

The paper is interesting and well written, but it has some major flaws that the authors themselves admit.

The authors are aware of the various limitations of the study and, in particular, the enormous prevalence of female subjects that makes it difficult to generalize the results.

Although the study has been a randomized, double-blind, placebo-controlled, parallel-group comparative study, the authors were unable to consider more health habits in their analyses.

The major deficiency, however, is the lack of diagnosis of infection:

Lines 90-93:“The subjects recorded their intake of tablets, the presence of infectious gastroenteritis, and their symptoms (fever (≥37 °C), abdominal pain, nausea, vomiting, and diarrhea) based on a diagnosis and subjective symptoms, and other physical changes using a self- completed questionnaires.”

What kind of infectious gastroenteritis?

If the authors are talking about viral gastroenteritis (since it is winter), what diagnosis was made?

It is well known that viral gastroenteritis is one of the most common illnesses in humans worldwide. As reported by Akihara et al. (2005), and cited by the authors, rotavirus, adenovirus, astrovirus, norovirus and sapovirus are considered to be significant global enteropathogens.

In particular, as a leading cause of severe diarrhoea in children, the pathogenic role of rotavirus in adults has been underestimated for a long time. Wang et al (2016) reported that, compared with bacterial pathogens, rotavirus infection from child-to-adult transmission is the most important epidemiologic setting generating a major impact on public health, i.e. increased adult burden of infectious gastroenteritis and genetic diversity of circulating rotaviruses. In this study it has been observed that adults infected with rotavirus developed more severe gastroenteritis symptoms accompanied with mild intestinal and blood inflammations. In this study rotavirus was detected in 48 of 441 (10.9%) specimens with prevalence peaking in December (33.3%) and January (27.9%).

In a study of Chikhi-Brachet et al. (2002), during one of the epidemic peak of acute diarrhoea observed each winter in France, calicivirus, rotavirus, adenovirus type 40 or 41, and astrovirus were detected in 16-65 years old sick individuals.

Onozuka et al (2019) reported that norovirus and rotavirus are two of the main causes of diarrhea in children in Japan (Thongprachum et al. 2015, 2016). Norovirus is a predominant winter virus, showing a marked peak in November and December with continual detection through the month of May. In contrast, rotavirus is most common in February and March (Thongprachum et al. 2015, 2016). Additionally, several studies conducted in Japan have reported that the overall prevalence of norovirus and rotavirus infection is 13.2–40.7% and 5.0–42.2%, respectively, with the number of detected cases increasing with each winter in Japan (Thongprachum et al. 2015, 2016).

Based on the literature it is therefore extremely important to understand what kind of "infectious gastroenteritis" the authors are talking about.

The data would be much more interesting if the type of pathogen sensitive to lactoferrin was known.

Akihara, S.; Phan, T.G.; Nguyen, T.A.; Hansman, G.; Okitsu, S.; Ushijima, H. Existence of multiple 289 outbreaks of viral gastroenteritis among infants in a day care center in Japan. Arch Virol 2005, 150, 2061-290 2075. doi: 10.1007/s00705-005-0540-y.

Chikhi-Brachet R, Bon F, Toubiana L, Pothier P, Nicolas JC, Flahault A, Kohli E. Virus diversity in a winter epidemic of acute diarrhea in France. J Clin Microbiol. 2002 Nov;40(11):4266-72. doi: 10.1128/jcm.40.11.4266-4272.2002. PMID: 12409408

Onozuka D, Gasparrini A, Sera F, Hashizume M, Honda Y. Modeling Future Projections of Temperature-Related Excess Morbidity due to Infectious Gastroenteritis under Climate Change Conditions in Japan. Environ Health Perspect. 2019 Jul;127(7):77006. doi: 10.1289/EHP4731. Epub 2019 Jul 19. PMID: 31322439; PMCID: PMC6792379.

Thongprachum A, Takanashi S, Kalesaran AF, Okitsu S, Mizuguchi M, Hayakawa S, Ushijima H Four-year study of viruses that cause diarrhea in Japanese pediatric outpatients. J Med Virol. 2015 Jul; 87(7):1141-8.

Thongprachum A, Khamrin P, Maneekarn N, Hayakawa S, Ushijima H Epidemiology of gastroenteritis viruses in Japan: Prevalence, seasonality, and outbreak.

J Med Virol. 2016 Apr; 88(4):551-70.

Wang Y, Zhang J, Liu P. Clinical and molecular epidemiologic trends reveal the important role of rotavirus in adult infectious gastroenteritis, in Shanghai, China. Infect Genet Evol. 2017 Jan;47:143-154. doi: 10.1016/j.meegid.2016.11.018. Epub 2016 Nov 22. PMID: 27884652.

Minor points

Some references, available only in Japanese, are not suitable for an international audience.

Reviewer 2 Report

The manuscript entitled "Effects of lactoferrin on prevention of infectious gastroenteritis in winter: A randomized, double-blinded, placebo-controlled trial for staff of kindergartens and nursery schools in Japan" written by Mizuiki et al is well presented.

Followings are the comments and suggestions for the authors:

Line 19: the reason for selecting winter season may be included here rather to wait until discussion section

Introduction section: need to be improved with relevance of the study carried out in winter season and also other control measures

Line 79: Trial carried out between Nov to March, Since you highlighted the winter season did u also take avg temp as consideration into your study ?

81: what is test foods ? Are these tablets with/out LF ?

Q) Most of the parameters are based on the qualitative measurements,Howe did you ensure the correct temperature measurement and also other symptoms were noted correctly ?

Fig 1: please explain well in legend or remove it, this is very confusing for the readers

Table 1: why female are more than male ?

Did you notice any side effects of the consumption of LF ?

Line 221 to 227 may be moved to introduction section

The limitations of the study such as very low male population and qualitative data entry seems to mislead the conclusion, how can you justify these things ?

Round 2

Reviewer 1 Report

Reviewers' comments:

This manuscript by dr. Mizuki and co-workers "Effects of lactoferrin on prevention of acute gastrointestinal symptoms in winter: A randomized, double-blinded, placebo-controlled trial for staff of kindergartens and nursery schools in Japan" examines the preventive effect of bovine lactoferrin on subjective acute gastrointestinal symptoms in childcare workers. The study was carried out in 335 healthy workers from 28 kindergartens and nursery schools during the winter season.

The revised paper is interesting, well structured, well written, and easy to follow. The title clearly indicates the focus of the paper and the Abstract section well summarizes the article contents. In the “Introduction” the context of the subject area is adequately addressed to justify the study and “Materials and methods” are appropriated and contain sufficient information for the research replication. The statistical analyses are well defined and appropriate to the design. Conclusion provides both interpretation of the results in the context of other evidence, and the implications for future research.

Minor points:

Abstract: lines 18-19:

This study investigated the preventive effects of lactoferrin (LF) on infectious gastroenteritis during the winter…..

Please substitute “infectious gastroenteritis” with “subjective acute gastrointestinal symptoms….”